# Status Forecasting Based on the Baseline Information Using Logistic Regresssion

**DOI:** 10.3390/e24101481

**Published:** 2022-10-17

**Authors:** Xin Zhao, Xiaokai Nie

**Affiliations:** 1School of Mathematics, Southeast University, Nanjing 210096, China; 2School of Automation, Southeast University, Nanjing 210096, China; 3Key Laboratory of Measurement and Control of Complex Systems of Engineering, Ministry of Education, Southeast University, Nanjing 210096, China; 4Shenzhen Research Institute, Southeast University, Shenzhen 518057, China

**Keywords:** status forecasting, subgroup analysis, baseline information, logistic regression

## Abstract

In the status forecasting problem, classification models such as logistic regression with input variables such as physiological, diagnostic, and treatment variables are typical ways of modeling. However, the parameter value and model performance differ among individuals with different baseline information. To cope with these difficulties, a subgroup analysis is conducted, in which models’ ANOVA and rpart are proposed to explore the influence of baseline information on the parameters and model performance. The results show that the logistic regression model achieves satisfactory performance, which is generally higher than 0.95 in AUC and around 0.9 in *F*1 and balanced accuracy. The subgroup analysis presents the prior parameter values for monitoring variables including SpO${}_{2}$, milrinone, non-opioid analgesics and dobutamine. The proposed method can be used to explore variables that are and are not medically related to the baseline variables.

## 1. Introduction

The intensive care unit, or ICU, is a hospital or medical center department that treats and manages patients with serious or life-threatening illnesses and injuries. Efficient real-time monitoring is the current method of patient care. However, such a monitoring system is not sufficient for status alerts beforehand, especially when danger statuses arrive suddenly. A status alert method is needed to support medical decisions. One of the effective methods for status alerting is to build a status forecasting model. The benefits of such a model are obvious. It can release the clinician from the burden of long-term nervousness. The clinician then can have more time for emergency preparation. In terms of the financial burden, such a method can decrease the costs for both patients and government. The saved costs can then be used for other purposes. The patients can also undergo less pain with the help of efficient medical care.

Status forecasting methods face many difficulties due to complex medical background. The response variable to be analyzed in such medical data can be continuous or categorical. In this research, the response variable that is analyzed is the status of the patient as either in danger or relatively safe. Compared to survival data, such a categorical variable is not censored, but the value changes between 0 and 1 until the patient is discharged from the ICU. Dynamically forecasting patient status is difficult between different patients. The baseline information of each patient, such as age, gender and weight, differs significantly, which may cause the model to have different performance and parameter values between different patients. It is important to analyze whether there is such a difference or not, and how the difference, if it exists, differs under different baseline information. Compared to a general model, building a status forecasting model under different baseline information can increase the accuracy of forecasting for a new patient, especially at the beginning of the forecast. The parameters can also be stored as the special character of such patient, thus reducing the data storage requirement in the context of streaming data.

In this research, a status forecasting method is developed for each patient using logistic regression, and a subgroup method is proposed for determining the prior parameter values of the model. The rest of the research is organized as follows. Section 2 presents a literature review of status forecasting methods, Section 3 introduces the proposed method, Section 4 demonstrates the results using real medical data, and Section 5 summarizes the work with a conclusion and prospects for further research. All computations were implemented using R [1].

## 2. Literature Review

Machine learning methods designed for status forecasting include neural networks, decision trees, support vector machines, and so on. Moor et al. [2] summarizes the machine learning research for sepsis early prediction. Subudhi et al. [3] compares machine learning methods for predicting ICU admission and mortality in COVID-19. Moghadam et al. [4] designs a machine learning algorithm to predict hypotension up to 30 min in advance based on the data from only 5 min of patient physiological history. Elhazmi et al. [5] uses the decision tree algorithm to predict mortality in critically ill adult COVID-19 patients admitted to the ICU. Rayan et al. [6] uses the support vector machine algorithm for sepsis prediction with good performance. In terms of medical support, such methods can be high in accuracy, but the interpretability maybe low due to the black box structure in most cases.

In the status forecasting problem, each variable can be regarded as a time series. Methods to analyze such time series include ARIMA, GARCH, and some deep learning methods such as LSTM and GAN. These methods can well describe a series’ trend or variation. Perone [7] compares the performance of ARIMA, ETS, NNAR, TBATS, and hybrid models to forecast the second wave of COVID-19 hospitalizations in Italy. Wei and Billings [8] proposes the nonlinear autoregressive moving average with an exogenous input model (NARMAX model includes the ARMA and ARIMA models as a special case). NARMAX has been most recently proposed and applied to modeling COVID-19 pandemic dynamics and understanding how weather conditions affect the spread of COVID-19. The model achieved good performance due to its transparent, interpretable, parsimonious, and simulatable properties. Caicedo-Torres and Gutierrez [9] develops visually interpretable deep learning for mortality prediction inside an ICU. Zhao et al. [10] develops an interval forecasting method for monitored variables in an ICU based on decision trees. The simulation and real data analysis show that the methods perform better than ARMA and GARCH. Che et al. [11] introduces a simple yet powerful gradient boosting tree method to learn interpretable models and, at the same time, achieve prediction performance as strong as deep learning models These models have good performance if the response variable is numeric, but may not perform as well for categorical variables.

Since the response variable is a categorical variable, logistic regression can be used, which models the probability of one event out of two alternatives. As logistic regression belongs to the category of generalized linear regression, the parameters can be used for subgroup analysis and stored to compress information. Ge et al. [12] uses logistic regression and recurrent neural networks to design an interpretable ICU mortality prediction method. Xu et al. [13] compares the performance of XGBoost and logistic regression in ICU mortality prediction in rheumatic heart disease. The results verify that the logistic regression model has convincing prediction. Vairavan et al. [14] uses logistic regression and a hidden Markov model to predict mortality in an ICU. Bennis et al. [15] utilizes the physiological cerebral parameters in a multivariable logistic regression model to improve prediction performance after 6 months for patients suffering brain injury. Logistic regression shows good performance in these applications. Therefore, it is proposed in this research.

The highlights of this research are that a subgroup analysis based on logistic regression is proposed. As an efficient status forecasting method, this analysis can provide valid parameter prior values for the model, as well as satisfy further purposes such as providing prior information for dynamical forecasting analysis of diseases. The novelties include that (1) a logistic regression model is used for categorical variables. The linear form of the model can also propose suitable parameter values for further research. (2) Instead of regarding all the patients as having the same data distribution, a subgroup analysis is conducted. (3) When new patients are admitted into the ICU, the prior distribution of the parameters can be given by the subgroup analysis.

## 3. Methods

For a typical individual *n*, the data for analysis can be divided into three parts. The first is multivariate time series $X_{n,\cdot ,\cdot }$, which are the input-monitored variables such as cardiac output, The data $X_{n,\cdot ,\cdot }$ can be described as
(1)$$X_{n,\;\cdot ,\;\cdot }=\left[\begin{array}{ccc} X_{n,1,1} & \cdots & X_{n,K,1}\\ X_{n,1,2} & \cdots & X_{n,K,2}\\ \vdots & \ddots & \vdots \\ X_{n,1,T_{n}} & \cdots & X_{n,K,T_{n}} \end{array}\right],$$
where $X_{n,\;\cdot ,\;\cdot }$ refers to the matrix that contains all monitored variables *K* at all times $T_{n}$. As the time length for individuals differs, $T_{n}$ is used to measure the time length for individual *n*. The other two parts are (2) the response variable $Y_{n,\cdot }$ and (3) the baseline variables $B_{n,\cdot }$, such as age, sex, and height.

The response variable $Y_{n,.}$ is defined as
(2)$$Y_{n,\cdot }={\left[y_{n,1},y_{n,2},\cdots ,y_{n,T_{n}}\right]}^{T},$$
where n=1,2,…,N and $t=1,2,\ldots ,T_{n}$. Generally, the $y_{n,t}$ is a categorical random variable with status values of 0 or 1.

### 3.1. Logistic Regression Model

In the logistic regression model, instead of referring to $Y_{n,\cdot }$ as the final response variable, the variable is transformed by the sigmoid function. The final model is expressed as
(3)$$log\frac{y_{n,t+1}}{1-y_{n,t+1}}=\left[X_{n,\cdot ,t},y_{n,t}\right]\theta_{n}+\epsilon_{n}$$
where $\theta_{n}={\left(\theta_{n,1},\theta_{n,2},\ldots ,\theta_{n,K+1}\right)}^{T}$ are unknown regression coefficients and $\epsilon_{n}$ are random errors with $E[\epsilon_{n}]=0$ and $\mathrm{Var}\left[\epsilon_{n}\right]=\sigma^{2}$. The coefficients $\theta_{n}$ differ across different individuals. The $X_{n,\cdot ,t}$ and $y_{n,t}$ at time *t* predict the $y_{n,t+1}$ at time t+1 with the first observation $y_{n,1}$ as known. After that, a stepwise method is applied to the model and only variables which have significant parameters are kept.

The model performance is measured by AUC (area under receiver operating characteristic (ROC)), *F*1, and balanced accuracy for each individual *n*. Suppose the confusion matrix result of a classification problem is that of Table 1.

The AUC value is the area under the receiver operating characteristic curve (ROC), which changes from 0 to 1. Generally, a model with an AUC of 0.5 has the same performance as random guessing, while a model with an AUC greater than 0.5 is suggested. The higher the AUC value, the better performance the model has.

The *F*1 score is defined as
(4)$$F1=\frac{2Precision*Recall}{Precision+Recall}$$
where *Precision* and *Recall* are
(5)$$Precision=\frac{TP}{TP+FP}$$
(6)$$Recall=\frac{TP}{TP+FN}.$$

The balanced accuracy is defined as
(7)$$BalancedAccuracy=\frac{TPR+TNR}{2}$$
where *TPR* and *TNR* are the true positive rate and true negative rate
(8)$$TPR=\frac{TP}{TP+FN}$$
(9)$$TNR=\frac{TN}{FP+TN}.$$

The model performance is measured by these three metrics.

### 3.2. Subgroup Analysis

In order to test whether the model performance is influenced by the baseline information or not, ANOVA is selected to test the differences of means using variance. The response variable is
(10)[AUC,F1,BalancedAccuracy]
and the input baseline variables are
(11)$$\left[\begin{array}{cccc} B_{1,1} & B_{1,2} & \cdots & B_{1,N_{B}}\\ B_{2,1} & B_{2,2} & \cdots & B_{2,N_{B}}\\ \cdots & \cdots & \ddots & \cdots \\ B_{N,1} & B_{N,2} & \cdots & B_{N,N_{B}} \end{array}\right].$$

In addition to the single variables, the interactive terms $\left\{B_{i}:B_{k}\right\}$ are also included in the ANOVA modeling, where $\left\{i,k\right\}\in \left\{1,2,\cdots ,N_{B}\right\}$ and i≠k.
(12)$$\left[AUC,\;F1,\;BalancedAccuracy\right]=f_{\mathrm{ANOVA}}^{k}\left\{B_{1},\;B_{2},\;\cdots ,\;B_{N_{B}},\;\left\{B_{i}:B_{k}\right\}\right\}.$$

The ANOVA model can show whether the baseline variables and their interactive terms have significant influence on the model performance or not.

After, in order to calculate the prior values for the significant variables in Equation (Equation 3), a recursive partitioning and regression trees (rpart) model is proposed to predict $\theta_{n}$ with the baseline variables. Instead of using all the observations across individuals 1 to *N* for $\theta_{k}$, only the observations across individuals 1 to $N_{s}$ (rearranged in order) whose parameters are significant for $\theta_{k}$ in the regression model (Equation (Equation 3)) are selected for modeling, with
(13)$${\left[\theta_{1,k},\theta_{2,k},\cdots ,\theta_{N_{s},k}\right]}^{T}$$
as the response variable. The input variables are
(14)$$\left[\begin{array}{cccc} B_{1,1} & B_{1,2} & \cdots & B_{1,N_{B}}\\ B_{2,1} & B_{2,2} & \cdots & B_{2,N_{B}}\\ \cdots & \cdots & \ddots & \cdots \\ B_{N_{s},1} & B_{N_{s},2} & \cdots & B_{N_{s},N_{B}} \end{array}\right].$$

These are observations are from the individuals whose $\theta_{k}$ is significant. The equation is
(15)$$\theta_{k}=f^{k}\left(B_{.,1},B_{.,2},\cdots ,B_{.,K_{B}}\right),$$
where $K_{B}$ is the number of baseline variables and k=1,2,⋯,K+1.

The model rpart works by iteratively choosing the most significant variable and its best split by using the criterion Gini gain. The stopping criterion is when the complexity parameter (cp) value reaches 0.005. The resulting decision rules can be used to propose the prior values. From this subgroup analysis, the most likely parameter values can be given based on the baseline information, which can offer valid prior information for the status forecasting model, thus improving the model robustness, especially at the beginning of the forecasting.

## 4. Real Data Analysis

This dataset [16] was collected during routine care at the Department of Intensive Care Medicine of the Bern University Hospital, Switzerland (ICU), an interdisciplinary 60-bed unit admitting more than 6500 patients per year. It was designed to study the early prediction of circulatory failure in the intensive care unit. The dataset in this research has been preprocessed by Hyland et al. [17], with outliers excluded and missing values imputed. The number of variables is 18, which include physiological variables, diagnostic test results, and treatment information, as shown in Table 2. For each patient, the observations range from hundreds to thousands; thus, a robust model can be built. To solve the problem that the statuses of some patients are all 0 or 1, and for some patients, none of the parameters are significant in Equation (Equation 3), the observations of those patients are deleted, including the patients whose status rarely changes, with standard deviation smaller than 0.1, and those with zero significant parameters. After, the number of patients in the analysis is 17,955.

From Figure 1, it is clear that the ratio of 0 to 1 differs across each baseline variable. In the subfigure of age, the value of age changes from 20 to 90, and the ratio of state 1 overall generally increases from 0.18 to 0.38 as age increases, which means individuals with higher age have a higher chance of being in a danger state. In the subfigure of height, the ratio differs little from around 0.3 when height is between 155 and 185, with some high and low values observed at high or low height. In the subfigure of sex, males have a higher ratio of 0.31 than the 0.27 of females, which means males are more likely to be in danger. In the subfigure of weight, individuals tend to have higher ratio of status 1 when weight increases from 60 kg to around 120kg, while this varies significantly afterwards.

In the subgroup analysis, the influence of baseline information on the model performance based on ANOVA was conducted, with the results shown in Figure 2. It can be seen that the relationships that are significant include those between sex and AUC, *F*1, and Accuracy; age and AUC; age and Accuracy; height and AUC; and age, weight, and AUC. Generally, most of the patients have good performance with AUC above 0.95, *F*1 above 0.9, and Accuracy around 0.9. It can be seen that observations belonging to males generally have better performance. In terms of age, the main black dots are around 40 to 80. In terms of weight and height, the main black dots are around 40 to 120 and 160 to 180.

Table 3 proposes the prior values for the variables, which play a significant role in individual status forecasting. The results are achieved using the model rpart by selecting the variables whose cp value is higher than 0.005. For SpO${}_{2}$, patients with higher weight tend to have higher parameter values. For milrinone, patients with higher age and higher weight tend to have parameters with higher values. For dobutamine, individuals with higher age have higher parameter values. For non-opioid analgesics, individuals who are males tend to have higher parameter values.

## 5. Conclusions

In this research, a logistic regression model is built to forecast the patient status and a subgroup analysis based on ANOVA and rpart is given to summarize the model information for individuals who share similar baseline information. The developed subgroup analysis generates the parameter results for each individual, which can be used as prior information for the logistic regression model and also compress massive streaming data into a few values. When the patients are admitted into the ICU, by referring to the baseline information, the logistic regression model can have better prior parameter values than the random suggested values. In that case, the model with the prior information can have better accuracy and stability than random guessing.

From the logistic regression results, the model performance differs among different baseline information according to the ANOVA results. For individuals with different characters, different suitable parameters should be suggested. The variables that are significant ro4 the model performance, measured by AUC, balanced accuracy, and *F*1, include sex, age, height, and some mutual effect such as the weight and age mutual effect on AUC. In the subgroup analysis, the prior parameter values are given according to the decision rules based on the baseline variables.

In future work, the subgroup analysis can take into account more variables such as patient history, medical images, and other pharmacy descriptions, in addition to the baseline information. A dynamic-streaming forecasting method can also be developed based on this prior parameter method. The logistic regression model in this research is linear regression, which can be easily modified compared to complex models such as machine-learning or deep-learning methods. However, it may have difficulty extracting the parameter information to propose the parameter prior distribution. In the ANOVA, mutual effect analysis based on two variables can be developed as three or more variables.

## Figures and Tables

**Figure 1 entropy-24-01481-f001:**
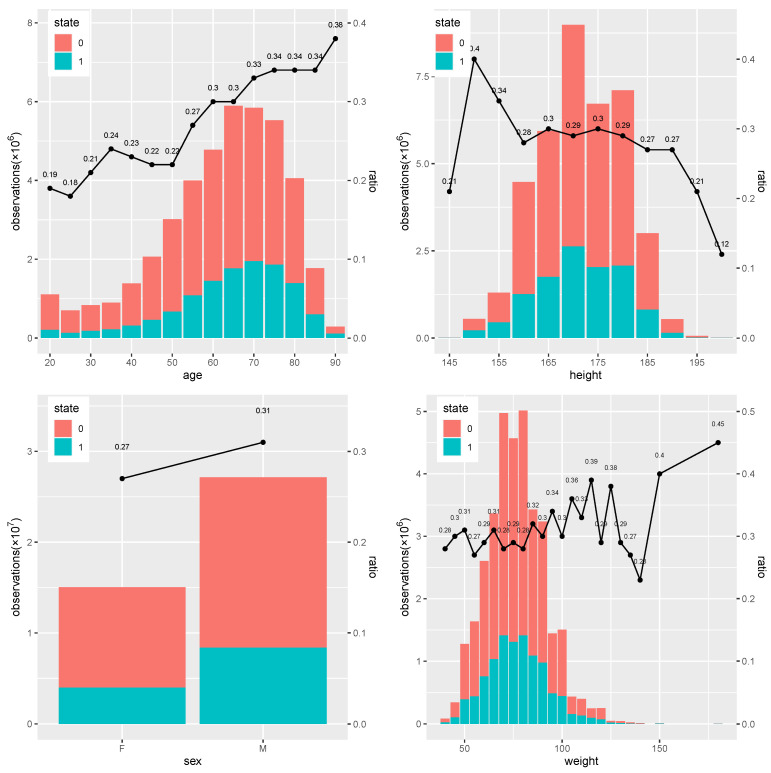
The baseline information under status 0 (safe) and 1 (danger), including age, height, sex, and weight. The ratio is the percentage of status 1 over all the observations across all the patients.

**Figure 2 entropy-24-01481-f002:**
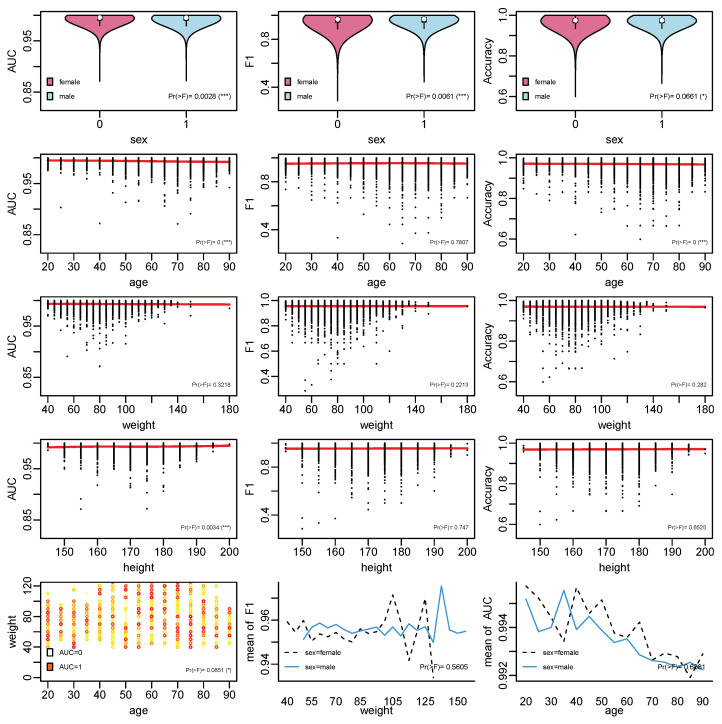
The results of ANOVA regression: the performance difference under different baseline information of sex, age, weight, and height measured by AUC, *F*1, and accuracy. The black dots are the results for each individual. Whether the difference is significant or not can be seen from the significant level, with * and *** representing significant levels of 0.1 and 0.01.

**Table 1 entropy-24-01481-t001:** The confusion matrix table of a classification problem.

	Actual Positive	Actual Negative
Predicted Positive	True Positive (TP)	False Positive (FP)
Predicted Negative	False Negative (FN)	True Negative (TN)

**Table 2 entropy-24-01481-t002:** The information of the 18 most predictive variables.

Variable	Unit
Heart rate	/min
Systolic BP (invasive)	mmHg
Diastolic BP (invasive)	mmHg
MAP	mmHg
Cardiac output	l/min
SpO${}_{2}$	%
RASS	10-point scale
Peak inspiratory pressure (ventilator)	cmH${}_{2}$O
Lactate arterial	mmol/L
Lactate venous	mmol/L
INR	ratio
Serum glucose	mmol/L
C-reactive protein	mg/L
Dobutamine	flow [mg/min]
Milrinone	flow [mg/min]
Levosimendan	flow [mg/min]
Theophyllin	flow [mg/min]
Non-opioid analgesics	binary indication of drug presence [yes/no]

**Table 3 entropy-24-01481-t003:** Prior parameter values of each variable if they have significant a relationship with the baseline information of age, sex, height, and weight.

Variable (Significant)	Unit	Decision Rules	Prior Parameter Value
SpO${}_{2}$	%	weight < 145	0.011
		weight ≥ 145	0.095
Milrinone	mg/kg/min	age ≥ 23	173
		age < 23 weight ≥ 145	83
		age < 23 weight < 145	1.8
Non-opioid analgesics	drug presence or not	sex = female	2
		sex = male	3.6
Dobutamine	mg/kg/min	age < 28	1.7
		28 ≤ age < 78	2.6
		age ≥ 78	5.4

## Data Availability

The source code in the method is available from the corresponding author upon request. The real data in application can be requested from Hyland et al. [17].

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
