# Peer review of "Status Forecasting Based on the Baseline Information Using Logistic Regression"

_entropy, 2022, doi:10.3390/e24101481_

Round 1

Reviewer 1 Report

Authors propose an application of Markov's models in the prediction of hospital stay length in patients. Good predictive models can help in the optimization of valuable resources in health management.

The paper introduces the problem and refers to some of the most well-known predictive models in the field, then focuses on Markov's, defining the problem of application.

The problem is clearly established and well described. Experiments are discussed and the application of model is well justified.

The paper is well written and very well structured, the reader should easily follow the document content.

Author Response

Thank you for your helpful review. We devoted more effort in improving the paper. We belive you will be more satisfied with it.  

Reviewer 2 Report

The authors present a Markov regression based method for forecasting patient status. However, I think the methodology is not described clearly/the design has serious flaws. Therefore, I cannot recommend it for publication in the current form.

1.       Starting from line 137, the formulation of the regression does not make sense. The transition status is not ordinal, so the transformation of y_{n,t}^m is not reasonable. In other words, there is no such ordinal relationship that (0->0) < (0->1) < (1->0) < (1->1). This problem is better formulated as a classification instead regression problem, and methods such as logistic regression (you can include the lagged status variable in predictors), and generalized Markov switching regression model (to estimate the transition matrix directly) are more suitable. The authors should redo the modeling, or describe their modeling approach in more detail if I misunderstood.

2.       The authors did not describe the data. A research article should be self-sufficient, and they cannot simply refer everything to ref [18] instead of make their own descriptions. I don’t even know what the 18 most predictive variables are.

3.       There is also no regression results shown. I would expect at least there will be a table showing the regression coefficients and significance.

4.       There are too many typos. Two lines above line 155, should be i!=k, not i!=j. Two lines above line 157, should be theta_k, not theta_n. Line 173, analized -> analyzed. Line 211, rule -> role. Please proofread carefully.

Author Response

  1. Starting from line 137, the formulation of the regression does not make sense. The transition status is not ordinal, so the transformation of y_{n,t}^m is not reasonable. In other words, there is no such ordinal relationship that (0->0) < (0->1) < (1->0) < (1->1). This problem is better formulated as a classification instead regression problem, and methods such as logistic regression (you can include the lagged status variable in predictors), and generalized Markov switching regression model (to estimate the transition matrix directly) are more suitable. The authors should redo the modeling, or describe their modeling approach in more detail if I misunderstood.

Thank you for your kindly suggestions. We add more detail in describing the ordinal variable y_{n,t}^m in the text as:

“Instead of regarding $y^m_{n,t}$ as discrete categorical variable, it is regarded as ordinal variable. The value of $y^m_{n,t}$ reveals the danger level of the patient. The higher the level, the more attention should be given to the patient. For example, level 4 means the patient keeps in danger while level 1 means the patient is currently out of danger.”

We also do the experiment by using logistic regression and find some difficulties in addressing the problem. For example, when the response variable is multiple, there are many formulas to be estimated and many more groups of parameters to be subgroup-analyzed which might decrease the prior distribution robustness based on the baseline information as the number of parameters increase a lot. If the response variable has two categories (status change or not), the model has not good performance due to the severe sample size imbalance. Due to those reasons, we think it might be good to be included in the future research paragraph as:

“If the response variable is regarded as discrete categorical variable instead of ordinal variable, some methods are suggested like logistic regression by including the lagged status variable in predictors, and generalized Markov switching regression model by estimating the transition matrix directly.”

  1. The authors did not describe the data. A research article should be self-sufficient, and they cannot simply refer everything to ref [18] instead of make their own descriptions. I don’t even know what the 18 most predictive variables are.

Thank you for your kindly comments. We add the information of the 18 most predictive variables in the text as Table 1. As the values of the variables differ across different patients, we do not add the statistical description in the text. If you prefer to add, we will do that.

  1. There is also no regression results shown. I would expect at least there will be a table showing the regression coefficients and significance.

Thank you for your helpful comments. A table with the results is truly helpful. But the experiment is done for each individual and there are 33905 individuals with more than 10 variables, so it will be a quite large table. Instead, we show the regression results in Figure 3 which shows how the $R^2$ and other metrics change under different baseline information. We will try better ways to show the results in the future research.

  1. There are too many typos. Two lines above line 155, should be i!=k, not i!=j. Two lines above line 157, should be theta_k, not theta_n. Line 173, analized -> analyzed. Line 211, rule -> role. Please proofread carefully.

Thank you for your helpful comments. The typos have been correctly.

Reviewer 3 Report

The paper applies a Markov regression model built on ANOVA and Ctree algorithms to ICU data modelling and prediction. The results obtained are interesting.  The paper is well organized and structured. The method used is clearly presented and evaluated through case study on real data. Experimental results are well reported.

A number of minor points and concerns are listed below. Hope these points and concerns can be well addressed in revision and hence help improve the quality of the paper.

1. It is recommended that the novelty of the proposed method be clearly indicated/highlighted in the revised paper. It is not obviously in the present version.

2.  Some equations are assigned equation numbers, but some other are not. For convenience of reading, it would be better to assign a number to each equation.

3.  In the Markov regression model, the values of the transformed variable in (between line 137 and line 138, and above Eq (1)) are defined to be 1, 2, 3, and 4 for the four modes. Why?

1) Can these values be assigned to be 0, 1, 2, 3 or -1, -2, -3, -4 or 10, 20, 30, 40?  Why (or why not)?

2) If the four values can be chosen to be other values rather than 1-4, how would it affect the experimental results reported in Section 4?

4.  Reading though the result from the linear regression model (shown in Figure 2), it is very clear that for all the 4 modes of the transformed variable, the errors between the prediction and real status is constant.       

Is the model that gives such a prediction result reasonable? An explanation is needed.

5.   The literature review can be enhanced by mentioning other models. For example, the nonlinear autoregressive moving average with exogenous input model (which includes ARMA and ARIMA models as a special case) has been proposed and applied to “modelling COVID-19 pandemic dynamics” and understanding “how weather conditions affect the spread of COVID-19” most recently (https://doi.org/10.1201/9781003176121). The most attractive properties of NARMAX model is that it is transparent, interpretable, parsimonious and simulatable (TIPS).

6. Typos:

1) Line 5, “model ANOVA and ” -- > the models ANOVA and

2) Line 16: “can only insist patients”. The meaning of the word ‘insist’ is unclear.

7. Line 4: "a Markov regression model is built to forecast the Markov transition state"

It may be better to make a change like: a Markov regression model is built to forecast [something else (but not its own state)]

The same for Line 230.

8. Line 72: “Xu et al. [2] compare”. A better alternative option may be to use past tense: Xu et al. [2] compared.

The same for many other cases, e.g., Lines 74, 76, 78, and so on throughout out the paper.

Author Response

A number of minor points and concerns are listed below. Hope these points and concerns can be well addressed in revision and hence help improve the quality of the paper.

  1. It is recommended that the novelty of the proposed method be clearly indicated/highlighted in the revised paper. It is not obviously in the present version.

Thank you for your helpful comments. The novelty has been updated at the end of Section 2.

“The highlight of this research is that a subgroup analysis based on Markov regression is proposed which can offer a status forecasting method and provide valid parameter prior value for the model as well as further purposes like being a prior information for dynamical forecasting analysis of diseases. The novelties include (1) the Markov states are chosen as the response variable instead of the original patient status. In most cases, clinicians are more concerned about whether the status changes or not. (2) A subgroup analysis is conducted instead of regarding all the patients having the same data distribution. (3) The prior distribution of the parameters are given by the subgroup analysis, which can increase the prediction performance when further patients are admitted into the hospital.”

  1. Some equations are assigned equation numbers, but some other are not. For convenience of reading, it would be better to assign a number to each equation.

Thank you for your suggestion. All the equations have their numbers added.

  1. In the Markov regression model, the values of the transformed variable in (between line 137 and line 138, and above Eq (1)) are defined to be 1, 2, 3, and 4 for the four modes. Why?

Thank you for your question. The four modes are originally four different Markov transition directions. They are further defined as 1,2,3 and 4 to reflect the danger level of the patient. We add more detail in the text to make it clearer:

“Instead of regarding $y^m_{n,t}$ as discrete categorical variable, it is regarded as ordinal variable. The value of $y^m_{n,t}$ reveals the danger level of the patient. The higher the level, the more attention should be given to the patient. For example, level 4 means the patient keeps in danger while level 1 means the patient is currently out of danger.”

1) Can these values be assigned to be 0, 1, 2, 3 or -1, -2, -3, -4 or 10, 20, 30, 40?  Why (or why not)?

These values can be assigned to other ordinal values like 0,1,2,3, as long as they can describe the danger level of the patient.

2) If the four values can be chosen to be other values rather than 1-4, how would it affect the experimental results reported in Section 4?

The results might change in the variable parameters like the corresponding change of the intercept. The significant level and model performance metrics like $R^2$ may not be reflected.

  1. Reading though the result from the linear regression model (shown in Figure 2), it is very clear that for all the 4 modes of the transformed variable, the errors between the prediction and real status is constant.       

Is the model that gives such a prediction result reasonable? An explanation is needed.

Thank you for your questions. The gap between the prediction and real status is not the errors. We add more detail in the caption of the Figure 2 as:

The result of linear regression with one individual as an example. The black dots are the real status with value 1 as status transforms from 0 to 0, and the rests are 2: 0-1; 3: 1-0; 4: 1-1. The red dots are the predicted status with one observation ahead. The corresponding values are state 1 if the predicted value belongs to (-$\infty$,1.5], and the rests are 2: (1.5, 2.5], 3: (2.5, 3.5], 4: (3.5, $\infty$) The red dots are the original values minus 0.2 so that the black and red lines will not override each other. Areas with only black or red dots are those misclassified.

  1. The literature review can be enhanced by mentioning other models. For example, the nonlinear autoregressive moving average with exogenous input model (which includes ARMA and ARIMA models as a special case) has been proposed and applied to “modelling COVID-19 pandemic dynamics” and understanding “how weather conditions affect the spread of COVID-19” most recently (https://doi.org/10.1201/9781003176121). The most attractive properties of NARMAX model is that it is transparent, interpretable, parsimonious and simulatable (TIPS).

Thank you for the reference. It has been added in Section 2 as:

“\citet{01763} proposed the nonlinear autoregressive moving average with exogenous input model (NARMAX model includes the ARMA and ARIMA models as a special case). NARMAX has been proposed and applied to modelling COVID-19 pandemic dynamics and understanding how weather conditions affect the spread of COVID-19 most recently. The model achieved good performance due to its transparent, interpretable, parsimonious and simulatable properties.”

  1. Typos:

1) Line 5, “model ANOVA and ” -- > the models ANOVA and

2) Line 16: “can only insist patients”. The meaning of the word ‘insist’ is unclear.

Thanks for the suggestions. The typos have been corrected.

  1. Line 4: "a Markov regression model is built to forecast the Markov transition state"

It may be better to make a change like: a Markov regression model is built to forecast [something else (but not its own state)]

The same for Line 230.

Thanks for the advice. “Markov transition state” has been changed to “patient danger level”.

  1. Line 72: “Xu et al. [2] compare”. A better alternative option may be to use past tense: Xu et al. [2] compared.

The same for many other cases, e.g., Lines 74, 76, 78, and so on throughout out the paper.

Thanks for your suggestions. They have been changed throughout the paper.

Round 2

Reviewer 2 Report

Thanks for the authors' response. However, I cannot agree with their argument on using ordinal responses. First, they need to justify the ordinal relationship. I can agree that (0->0) < (1->1), but why (0->1) < (1->0)? Shouldn't the patient going to be in status 1 be in higher danger level than a recovering patient? Second, they need to justify that the distance between each status are the same, e.g. the distance (0->1) - (0->0) = (1->1) - (1->0) and so on.

 They claim that there're difficulties using logistic regression because of multiple response variables. Can they do it as a binary classification, where response is next status, and include all the features and current status in the predictors? You still only need one equation per patient. I don't think it's necessary to use the 4 transitions or whether the status is changed as the response.

They authors claims that they fit one model for each patient. Please also justify why this is necessary. It is very likely to overfit this way. Are the fitted model across individuals very different? This approach also brings limitations to new patients where data must be collected first to fit a model.

Author Response

Thank you for your suggestions. We have changed the Markov regression to Logistic regression, where the response is next status, and all the features and the current status are included in the predictors.

Fitting one model for each patient is due to the reason that subgroup analysis is conducted to test whether the performance and parameters have significant different among different baseline variables. The results show that there are truly significant differences among different baseline variables. As such baseline information like age, sex, height and weight have been stored in the medical system generally, which are the instant information that can be analyzed when a new patient is admitted, in that case this information does not need to be extra collected as they have already existed in the medical system. That is also why this information can be used for prior values suggestion.

Reviewer 3 Report

I'd like to thank authors for their good effort in revising and significantly improving the paper. 

No further comment.

Author Response

Thank you very much for your helpful suggestions.

Round 3

Reviewer 2 Report

Thanks for the effort in revising the manuscript. I do not have further comments and recommend it for publication.